# Atmospheric-moisture-induced polyacrylate hydrogels for hybrid passive cooling

Roisul Hasan Galib[1,6], Yanpei Tian[2,3,6], Yue Lei[3,4], Saichao Dang[2,3], Xiaole Li[2,5], Arief Yudhanto[2,5], Gilles Lubineau [2,5] & Qiaoqiang Gan [1,2,3] ✉

Heat stress is being exacerbated by global warming, jeopardizing human and social sustainability. As a result, reliable and energy-efficient cooling methods are highly sought-after. Here, we report a polyacrylate film fabricated by self-moisture-absorbing hygroscopic hydrogel for efficient hybrid passive cooling. Using one of the lowest-cost industrial materials (e.g., sodium polyacrylate), we demonstrate radiative cooling by reducing solar heating with high solar reflectance (0.93) while maximizing thermal emission with high mid-infrared emittance (0.99). Importantly, the manufacturing process utilizes only atmospheric moisture and requires no additional chemicals or energy consumption, making it a completely green process. Under sunlight illumination of 800 W m$^{-2}$, the surface temperature of the film was reduced by 5 °C under a partly cloudy sky observed at Buffalo, NY. Combined with its hygroscopic feature, this film can simultaneously introduce evaporative cooling that is independent of access to the clear sky. The hybrid passive cooling approach is projected to decrease global carbon emissions by 118.4 billion kg/year compared to current air-conditioning facilities powered by electricity. Given its low-cost raw materials and excellent molding feature, the film can be manufactured through simple and cost-effective roll-to-roll processes, making it suitable for future building construction and personal thermal management needs.

Unprecedented and extreme heat waves have recently scorched northern hemisphere swathes and threatened humans' survival in overheated areas[1]. This extreme heatwave is a lethal consequence of global warming, with more frequent strikes and higher temperatures expected in the future[2,3]. As a result, rapidly increasing energy demands from air conditioning systems are causing frequent power outages and overloading energy grids[4–6]. Energy consumption and associated environmental issues by compressor-based cooling techniques amplify the need to explore the energy-free passive cooling approach[7–9]. The natural process of evaporation,

condensation, precipitation, and percolation, known as the "water cycle", delineates the continuous movement of water between the Earth and the atmosphere in three phases[10]. Particularly, a large amount of energy is absorbed during the water evaporation process (e.g., 2,430 kJ kg$^{-1}$ at 30 °C). Therefore, it is a viable cooling approach of evaporation for passively absorbing ambient heat[11–13]. However, continuous evaporative cooling requires an external water supply, limiting its widespread use in areas with limited water sources[14,15]. Global atmospheric vapor is estimated to be 12,900 km$^3$ and is considered a ubiquitous water resource[16,17]. Moreover, rising ambient

[1]Department of Electrical Engineering, University at Buffalo, The State University of New York, Buffalo, NY 14260, USA. [2]Division of Physical Science and Engineering, King Abdullah University of Science and Technology (KAUST), Thuwal 23955-6900, Saudi Arabia. [3]Water Desalination and Reuse Center, Biological and Environmental Science & Engineering Division, KAUST, Thuwal 23955-6900, Saudi Arabia. [4]School of Architecture and Urban Planning, Chongqing University, 400045 Chongqing, China. [5]Mechanics of Composites for Energy and Mobility Laboratory, KAUST, Thuwal 23955-6900, Saudi Arabia. [6]These authors contributed equally: Roisul Hasan Galib, Yanpei Tian. ✉e-mail: qiaoqiang.gan@kaust.edu.sa

temperatures cause an increase in atmospheric moisture, amplifying the warming effect of greenhouse gases such as moisture (moisture is the largest component of greenhouse gases and even more challenging to control than carbon dioxide[18,19]). Therefore, utilizing atmospheric moisture effectively could create a solution to address the challenge of global warming.

Desiccant materials, such as silica gel[20], zeolites[21], hygroscopic hydrogel[18], metal-organic-frameworks[22,23], and lithium chloride[24], have been extensively investigated for absorption of vapor from humid atmospheres during the nighttime and evaporation during the daytime for water harvesting needs[25]. Among these, the hygroscopic hydrogel has been developed for atmospheric water harvesting[18,19,26], evaporative cooling[25,27], and interfacial solar evaporation[28,29] due to its high affinity to absorb water and relatively low temperature-activated desorption process (~ 30 °C). The inherent molecular vibrations of chemical bonds over infrared wavelengths (5–25 μm) also endow hygroscopic hydrogels with high thermal emittance for effective heat dissipation through radiative cooling[30,31]. Recently, bilayer structures with polymer fibrous networks atop a hygroscopic hydrogel underlayer have been evaluated for the integration of radiative and evaporative cooling[25,32]. Representative hygroscopic hydrogel includes poly(N-isopropylacrylamide) (pNIPAM), Polyvinyl alcohol (PVA), and polyacrylamide (PAM) have also been reported as the hygroscopic matrix, while LiCl and LiBr salts serve to enhance moisture absorption for enhanced cooling performance (see Supplementary Table 1 for comparison). However, these hydrogel layers require a complex synthesis process, which boosts the fabrication cost and hinders its extensive engineering implementations. Specifically, the fabrication of poly (vinyl alcohol)·CaCl2 hydrogel involves time-sensitive and labor-intensive processes such as gelation, freeze-drying, and moisture-absorbent loading. These processes contribute to positive carbon emissions and exacerbate global warming. Therefore, a completely "green" fabrication process with no carbon emissions is highly desired to develop the hybrid passive cooling technique.

In this article, we propose an approach for hybrid radiative and evaporative cooling by transforming sodium polyacrylate (PAAS) powder into a white and continuous film using atmospheric moisture only, creating an environmentally friendly photonic hydrogel with a carbon-free manufacturing process. The porous structure of the photonic hydrogel allows efficient sunlight reflection, which reduces solar heating. Additionally, the intrinsic molecular vibrations of PAAS polymer chains of the hydrogel provide a high mid-infrared emittance, which expedites radiative heat dissipation through the atmospheric window. Simultaneously, hydrogel can store water harvested from the atmosphere at night which can be employed for daytime evaporative cooling to enhance its cooling effect. Under a partly cloudy sky with a solar intensity of 800 W m⁻², a subambient temperature reduction of 5 °C (from 37 °C to 32 °C) was achieved due to the hybrid cooling mechanism. This performance is superior to conventional radiative cooling systems (i.e., the pure radiative cooling sample even climbed 2 °C above the ambient temperature under identical environmental conditions). Intriguingly, the film can be recycled through crushing and re-crosslinking (i.e., self-healing), which maintains its optical performance, prolongs its lifespan, and hence decreases the lifespan cost. The hybrid passive cooling feature of this PAAS photonic film expands its potential applications in conditions of weakened radiative cooling in high-humid weather[33] or in urban areas where access to the open sky is restricted by high-rise buildings[34]. Importantly, this moisture-absorption-induced PAAS hydrogel facilitates a micro-water cycle for passive cooling, representing a prospective usage of atmospheric moisture. Combined with its environmental sustainability during the manufacturing processes, the implementation of this passive cooling film will result in net negative carbon emissions in the long-term deployment.

## Results
### Working mechanism and fabrication of the mechanically robust PAAS photonic film

Sodium polyacrylate (PAAS), also called water lock, is a superabsorbent polymer made of repeated chains of acrylate compounds[35] with many other favorable features, including mechanical durability and excellent thermal resistance[36]. Recently, renewed interest in this anionic polyelectrolyte material has emerged due to new applications in expansion microscopy for bioimaging[37,38] and pressure-sensitive adhesives[39]. Generally, it is commercially available in the form of white powders due to the backscattering of ambient light and has been utilized in many daily commercial products, like baby diapers. These white particles can strongly backscatter incident sunlight, which is a desirable property for radiative cooling. However, the powder form in a dry state makes PAAS inconvenient for many practical applications like building envelopes and personal thermal management. Here, we report a process to fabricate PAAS photonic film and expand its application for hybrid passive cooling. Figure 1 illustrates the working principle, scalable fabrication, and mechanical robustness of this PAAS photonic film. The porous PAAS photonic film can absorb moisture from its surroundings, particularly in areas with high relative humidity at nighttime (as illustrated in Fig. 1a). The harvested water evaporates and carries the heat away from the film upon heating under the illumination of sunlight during the daytime period. In this situation, the porous structure backscatters sunlight to reduce the solar heating effect. Simultaneously, the heat is dissipated to outer space through the atmospheric window in the form of thermal radiation. As a result, it can achieve hybrid passive radiative cooling due to its highly scattering feature and high thermal emittance from the molecular vibrations of the PAAS polymer chains. Intriguingly, this PAAS hydrogel can regenerate itself by absorbing moisture when the ambient humidity increases (e.g., at nighttime). The evaporation and self-adsorption processes form a perfect daily water cycle that facilitates the heat absorption and release process synchronously. More importantly, the entire manufacturing is amenable to industrial roll-to-roll processes (including PAAS dry powder feeding, moisture-induced crosslinking, and natural air drying as illustrated in Fig. 1b and Supplementary Note 1, Supplementary Fig. 1). Figure 1c displays the transformation process of PAAS photonic film. It starts from the raw material of white PAAS dry powder with a diameter of around 100 μm (i.e., a layer of PAAS powder of about 1 mm, see the SEM image of PAAS dry powder in Supplementary Fig. 2) that is widely used as the inclusion of baby diapers. By spraying moisture over its top surface and swelling, it is finally transfigured into a free-standing white film with dimensions of 1 m × 0.4 m × 3 mm (see the lower panel in Fig. 1c). Remarkably, this PAAS photonic film can even be fabricated by the atmospheric moisture-induced process without artificially spraying water. It involves a simple process of spreading the dry PAAS powders (thickness, ~2 mm) on a flat substrate in an outdoor environment with a high relativity humidity from 65% to 90% at night in Buffalo, NY. A continuous film can be generated within 6 h during the night (Supplementary Fig. 3). According to our experiment under the ambient temperature of around 22°C, the minimum relative humidity for dry PAAS powders to form a continuous film is 60%, as illustrated in Supplementary Fig. 4. This "natural" and "green" method further simplifies the fabrication process, minimizing the labor and energy requirement for large-scale deployment. Moreover, color pigments can also be introduced to form a colorful PAAS photonic film without affecting the formation of the continuous film (see Supplementary Fig. 5), which increases the versatility of PAAS photonic film for aesthetic purposes. In addition, the PAAS photonic film can also be applied onto diverse substrates with different textures like plastics, wood, and metals, which broadens its application scenarios like roofing, outdoor electronics, automotive, and industrial (see Supplementary Note 2, Supplementary Fig. 6).

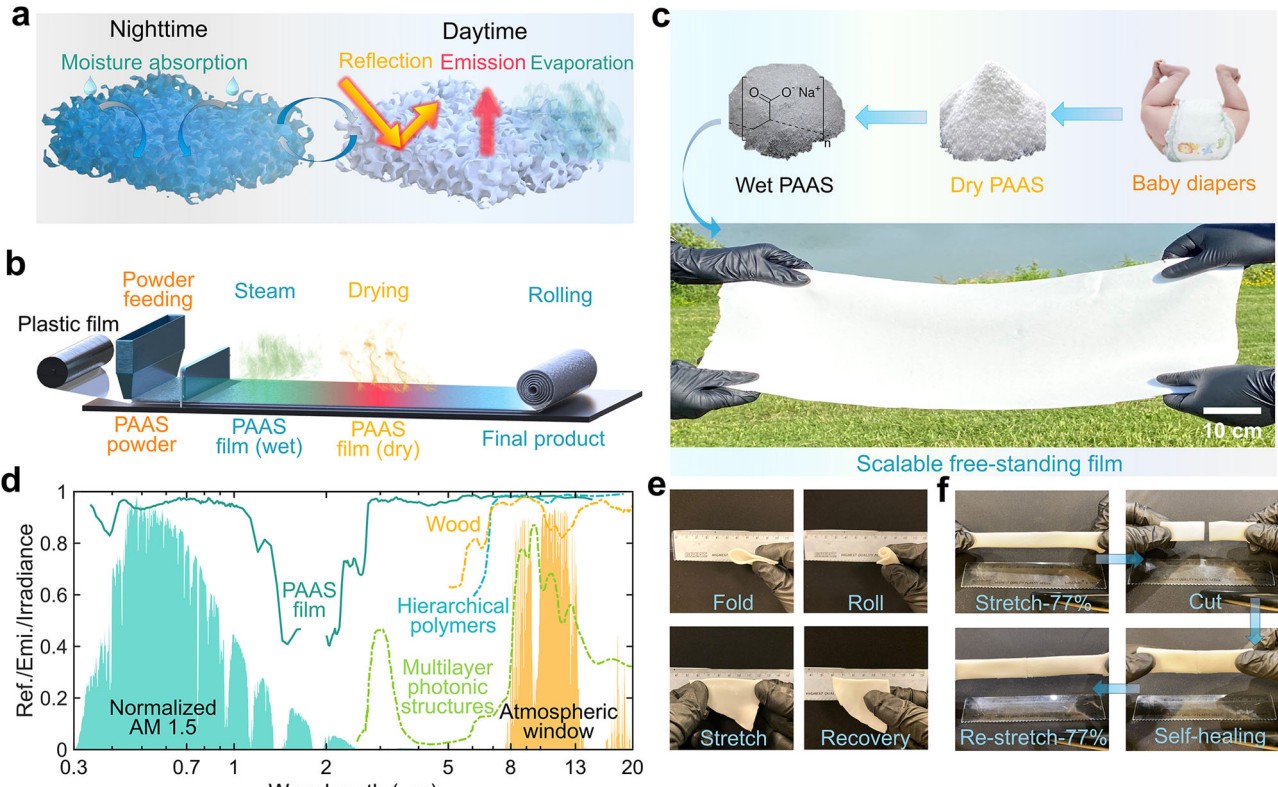

**Fig. 1 | Fabrication, working principle, spectra, and mechanical properties of sodium polyacrylate (PAAS) photonic film. a** The cooling mechanism of the passive radiative and evaporative PAAS photonic film. It absorbs atmospheric moisture during the nighttime and evaporates under direct sunlight. The porous structures backscatter the sunlight, and the infrared molecular vibrations enable high mid-infrared emittance for efficient radiation heat dissipation. **b** The roll-to-roll fabrication of PAAS photonic film is assisted by moisture-induced crosslinking and natural drying processes. **c** The transfiguring procedures of the mechanical-robust and free-standing PAAS photonic film from the regular commercial inclusions (PAAS dry powders), generally used as the baby diaper filler, to radiative or evaporative cooling films. **d** The reflectance and emittance spectra of the PAAS photonic film compared with the cooling wood[42], hierarchical polymers[43], and multilayer photonic structures[41], displaying against the standard solar irradiance spectrum (ASTM G173, global tilt) and the transmittance spectrum of the atmospheric window. **e** Photos demonstrating recovery of the PAAS photonic film after being folded, rolled, and stretched. **f** The flexibility of PAAS photonic film upon stretching; the re-stretching after cutting is possible due to the self-healing capability of PAAS photonic film.

Apart from its scalable and facial fabrication technique, the free-standing PAAS photonic film also shows excellent whiteness with a solar reflectance of 0.93 in its dry state, as shown in Fig. 1d (solid curves). The strong solar reflectance results from the efficient back-scattering feature of porous structures, as exemplified by the scattering effect of the red laser spot on the top surface of the film (Supplementary Fig. 7). The scattering area that shines on the PAAS photonic film is twice the size of white printing paper, which owes its whiteness to the backscattering of ambient light by the hierarchical cellulose fibers[40]. Importantly, the PAAS photonic film displays a thermal emittance of 0.99 over the atmospheric window (see the solid green curve over the 8–13 μm region in Fig. 1d), which is higher than multilayered photonic structures[41] and cooling wood[42] and is comparable to the hierarchical polymers[43] (see dashed curves in Fig. 1d).

It should be noted that the general photonic structures require complicated nanofabrication equipment[44,45]. For instance, cooling wood relies on the bleaching process to remove the lignin from the pristine wood and the heat-pressing process to form a densified form of microstructures. Here, the bleaching process requires the use of chemicals, such as sodium hydroxide, sodium sulfite, or hydrogen peroxide. Meanwhile, the subsequent overnight heat-pressing process also requires extra energy consumption and an extended processing time (e.g., overnight)[42]. Moreover, the phase inversion process for obtaining the hierarchical polymer films involves the utilization of a large quantity of volatile and harmful organic solvents (e.g., acetone[46]). In contrast, our environment-friendly fabrication method for producing PAAS photonic films using atmospheric moisture and cost-effective raw materials demonstrates the advantage of an industrial-level "green" manufacturing process. Remarkably, our PAAS photonic film with a thickness of 2 mm manufactured using this "green" process exhibits excellent flexibility, as demonstrated by its ability to quickly recover after being severely folded, rolled, or stretched (Fig. 1e). In addition, the PAAS photonic film has the capability of being stretched to approximately 1.8 times of its original length (as illustrated in Fig. 1f) before a tearing occurs. Interestingly, after such a tearing (Supplementary Fig. 8), the PAAS photonic film could be reconnected as a result of the healing process by simply applying 90–95% relative humidity for 6 h (overnight). The "healed" PAAS photonic film can be stretched up to 1.5 times without experiencing a mechanical failure. The mechanical flexibility and self-healing ability of the PAAS photonic films potentially reduce the maintenance cost of numerous outdoor applications, particularly in radiative or evaporative cooling systems.

In addition to the mechanical and self-healing advantages, PAAS photonic film can be recycled easily using the usual steps of freezing the used film with liquid nitrogen, crushing it into powders, and forming the recycled film by moisturizing the mixture of crushed powders with pristine PAAS powders (see Supplementary Fig. 9 for the fabrication details). The recycled PAAS photonic film has a reduction of 0.02 in solar reflectance while keeping its original mid-infrared thermal emittance, extending its lifetime without significantly sacrificing the optical performance (Supplementary Fig. 10). Overall, the cost-effective material (~ \$1.2 m$^{-2}$, see Supplementary Table 2), high-

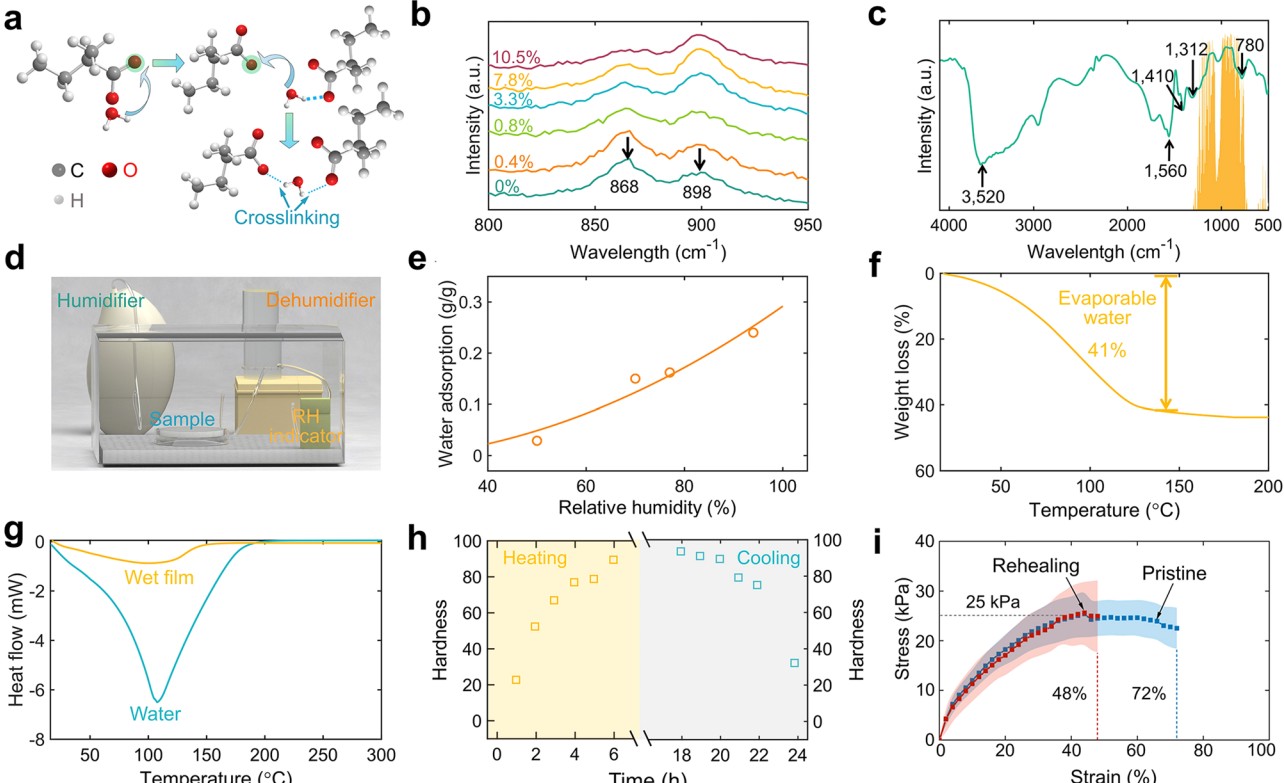

**Fig. 2 | Materials characterization. a** Schematic showing the sodium polyacrylate (PAAS) photonic film formation mechanism by hydrogel bond crosslink. **b** Evolution of Raman spectra of PAAS with the increasing water content. **c** FTIR spectra of PAAS powders displaying the chemical groups over the atmospheric window (solid light green area). **d** The experimental setup for the water content measurement under controlled air humidity and temperature. **e** Water absorption rates of PAAS powders at various relative humidity. **f, g** The TGA (**f**) and DSC (**g**) curves of PAAS photonic film and water. **h** The hardness variations of PAAS photonic film under solar heating and nocturnal cooling scenarios. **i** Stress–strain responses of PAAS photonic films.

yield manufacturing process, mechanical robustness, and self-healing capability of the PAAS photonic film render it a promising candidate for passive cooling applications on large scales. Next, we will perform systematic characterization and simulation to explore the physical mechanism of the unique optothermal properties of the PAAS photonic film.

## Material characterization of the PAAS photonic film

Figure 2a depicts the mechanism behind the formation of a mechanically robust continuous film from friable PAAS powder. The long chains of PAAS polymer are initially coiled up within the dry powder. When the dried chains absorb water, they stretch out due to the presence of a negative charge within each repeating unit of PAAS[47]: i.e., as water is added to the polymer, these areas form negatively charged ions that repel one another and cause the polymer to stretch out. Observation under an optical microscope reveals that, in the presence of moisture, the PAAS powders swell, come into contact with each other, and ultimately form a continuous film (Supplementary Fig. 11). When adjacent powders connect during swelling, hydrogel bonds form between different PAAS molecules, forming a crosslinked network. As shown in Fig. 2b, the Raman spectra explain the formation of hydrogen bonds between adjacent PAAS powders with different water contents. The peaks at 868 cm$^{-1}$ and 898 cm$^{-1}$ are attributable to the C−COO− stretching vibrations and the out-of-plane bending in the −OH of C−COOH, respectively. With the increasing water content, the peak at 868 cm$^{-1}$ weakens while the peak at 898 cm$^{-1}$ increases because the degree of ionization of PAAS increases[48]. The Fourier transform infrared spectrum (FTIR) elucidates the corresponding molecular vibrations of chemical bonds for the moisturized PAAS overlapping with the atmospheric a

3520 cm$^{-1}$ denotes the H−O−H bond stretching vibration of interlayer water molecules of PAAS after absorbing water. The peak at 1410 cm$^{-1}$ is ascribed to the −(CH2)$_n$− plane rocking vibrations and carbonates. The peak at 1560 cm$^{-1}$ corresponds to the carboxylate (COO−) radical vibration and C＝H asymmetric stretching vibration[49]. The −OH of PAAS gives it a unique feature of moisture adsorption in high-humidity conditions which is essential for evaporative cooling, while chemical groups of −(CH2)$_n$− and (COO−) offer high thermal emittance over the atmospheric window.

To investigate the water absorption capability of the PAAS photonic film, a piece of PAAS sample is placed into a humidity- and temperature-adjustable chamber (34 cm × 24 cm × 20 cm, Fig. 2d, and Supplementary Fig. 12) to record its mass change over the relative humidity. The humidity inside the chamber is adjusted by intermittent work of a humidifier and dehumidifier controlled by a proportional–integral–derivative (PID) controller. The water absorption rate of PAAS photonic film ranges from 0.05 g g$^{-1}$ (relative humidity, 50%) to 0.27 g g$^{-1}$ (relative humidity, 90%) (Fig. 2e). Thermogravimetry (TGA) characterization manifests that the evaporation water in PAAS photonic film is around 41% of its total mass, making it possible for evaporative cooling (Fig. 2f). Heating the PAAS photonic film above 200 °C would cause a significant color change that is followed by burning into char, demonstrating its excellent optical and thermal tolerance (Supplementary Fig. 13). Differential scanning calorimetric (DSC) elucidates the evaporation enthalpy of water in the PAAS hydrogel network (Fig. 2g). The DSC thermogram indicates a sharp peak at 100 °C, which is followed by a rapid decrease in heat flux, implying quick water evaporation. Notably, the heat flux signal peak for the PAAS photonic film is lower than that of pure water, demonstrating that the film provides a

larger surface area for water evaporation, thus enhancing its evaporative cooling capability.

Adding to their great potential towards heat, our developed PAAS photonic films also show fascinating mechanical characteristics in the superseding of the day-and-night cycle. During such a cycle, where the solar intensity and relative humidity are continuously changing, the mechanical properties of PAAS films are affected by varying the water content. To simulate the process, heating, and cooling cycles are mimicked in the PAAS photonic films (Fig. 2h). The heating step is performed by employing a solar intensity of 1000 W m$^{-2}$, a temperature of 31.4 °C, and relative humidity of 17%. The nocturnal cooling step is realized by keeping the PAAS photonic film in a dark chamber (zero solar intensity) at a temperature of 23.2 °C and relative humidity of 86%. The hardness of the photonic film is then measured at each stage using a durometer with scale A. Figure 2h shows that the hardness of the PAAS photonic film increased and peaked after 6 h of heating, indicating the considerably slow evaporation of water from the film. Upon cooling, the hardness of PAAS photonic film decreased as it absorbed water from the environment, indicating the ability of the film to recover its flexibility.

Furthermore, to demonstrate the flexibility and self-healing capability quantitatively, we perform tensile tests on "pristine" and "cutting and healing" PAAS specimens. Here, "pristine" means that the specimens are subjected to a relativity humidity of 90–95% at 25 °C, while "cut and rehealing" means that the specimens are severed into two pieces, reconnected, and subjected to relativity humidity of 90–95% at 25 °C for 6 h. The details of mechanical tests are given in Supplementary Note 3 (Supplementary Fig. 14). The stress−strain curves displayed in Fig. 2i show that the "pristine" PAAS photonic films exhibit strength of 25 kPa and failure strain of 72%. The rehealing process using ambient temperature with relativity humidity of 90−95% can completely recover the strength of the PAAS photonic film and most of its ductility (failure strain of 48%). It is expected that rehealing the severed specimens for a longer period (>24 h) may be able to recover the stretchability completely. Nonetheless, we demonstrate that there is a great potential for the hydrogel to completely heal itself after being severed or cut. The implication of "drying" on the mechanical response of PAAS photonic film is studied by leaving the film at an ambient temperature of 25 °C and relativity humidity of 55% for 60 days. Supplementary Fig. 15 shows that the drying process can improve the strength and ductility of up to 2 times. In addition, Supplementary Fig. 16 shows the potential of PAAS hydrogels for shape- and volume-changing materials due to their compressibility. Next, we show the characterization of optothermal features of PAAS photonic film to further reveal its potential for radiative cooling.

## Optical and thermal characterization of the PAAS photonic film

The PAAS photonic film appears as a diffused white as seen by the naked eye (i.e., Fig. 1c). As illustrated by the angle-dependent solar reflectance measurement from 0° to 60° in Fig. 3a, its solar reflectance is over 0.93, resulting in minimal solar heating under direct sunlight. Compared with a mirror-like radiative cooling surface[41,50], the high solar reflectance is mostly attributed to its hierarchical architected structures. The micro-sized PAAS structures coexist with the nanostructured surface textures of hierarchical-architected porous structures, evident from the scanning electron microscope (SEM) images (Fig. 3b, Supplementary Note 4, and Supplementary Figs. S17−S19). As the water content in the PAAS photonic film increases, the visible reflectance diminishes due to the decrease in refractive index between PAAS and ambient air. Simultaneously, higher water content leads to a noticeable disparity across infrared wavelengths (Supplementary Fig. 20). As shown in Fig. 3c, the majority of microstructures are ranging from 10 μm to 50 μm and 100 nm to 600 nm, respectively. As described by the refractive index in Fig. 3d extracted by the fitting of transmittance spectra (see technical details in Supplementary Note 4,

Supplementary Fig. 21), the PAAS shows a negligible extinction coefficient, $k$. As a result, its micro- and nanostructures efficiently backscatter longer (>500 nm) and shorter wavelengths (300−500 nm) of the sunlight, respectively, as validated by the scattering efficiency analysis (Fig. 3e). The abundant micro- or nanostructures can efficiently backscatter sunlight of solar wavelengths, where nanostructures reflect visible wavelengths and microstructures reflect visible and near-infrared wavelengths of sunlight. To further reveal the photon scattering phenomenon, finite-difference time-domain (FDTD) simulations are conducted for four characteristic wavelengths (300 nm, 500 nm, 1000 nm, and 1700 nm) within the solar spectrum (Fig. 3f). It can be observed that the propagation of long wavelengths (i.e., $\lambda = 1000$ and 1700 nm) penetrates much deeper into the surface compared to that at shorter wavelengths (i.e., $\lambda = 300$ and 500 nm). Therefore, the PAAS porous structure is more efficient in scattering the photons at shorter wavelengths, which is also correlated with the solar reflectance reduction in Fig. 1d (see the reduced solar reflectance over near-infrared wavelengths from 1.0 μm to 2.5 μm). When the PAAS photonic film is moisturized, it exhibits a reduced scattering effect. This can be attributed to the smaller refractive index difference between the moisturized PAAS and air, compared to the difference between dry PAAS and air. Specifically, the absorption of water by the PAAS photonic film results in a decrease in its refractive index, as the refractive indices of PAAS, water, and air are 1.49, 1.33, and 1.0, respectively, at a wavelength of 500 nm. The reduced scattering is also exemplified by the reduced solar reflectance of moisturized PAAS photonic film (Supplementary Fig. 22), where the moisturized PAAS photonic film displays a reduced solar reflectance of 0.89. Compared with the dry PAAS surface, the O−H stretching vibration of absorbed water in the moisturized PAAS photonic film induces a higher thermal emittance for more efficient radiative heat dissipation.

Quantifying the water evaporation behavior of the moisturized PAAS photonic film under varying solar intensities and different humilities helps understand its cooling potential. The evaporation rate of moisturized PAAS photonic film is measured as a function of solar intensity and humidity (Fig. 3g). A related theoretical model of the radiative and evaporative cooling performance for the PAAS photonic film is presented in Supplementary Note 5. As shown in Fig. 3g, indoor simulated evaporation tests reveal that water evaporation is promoted when the solar irradiance and wind speed are increased, or the ambient humidity is reduced. The evaporation rate reaches 1.75 kg m$^{-2}$ h$^{-1}$ at $23 \pm 0.5$ °C, relativity humidity of $20 \pm 2\%$, and a wind speed of 1 m s$^{-1}$. Under windy conditions, convection is the dominant factor for water evaporation, as the evaporation rate shows a slow increase with a change in solar irradiance intensity from 0 kW m$^{-2}$ to 1.0 kW m$^{-2}$ (see the blue line in Fig. 3g). While there is no wind and the relativity humidity is high (e.g., relativity humidity is 60% and wind speed is 0 m s$^{-1}$), the solar heating-enhanced evaporation is dominant since the evaporation rate increases quickly from 0.12 kg m$^{-2}$ h$^{-1}$ to 0.43 kg m$^{-2}$ h$^{-1}$ with the increase of solar intensity from 0 kW m$^{-2}$ to 1.0 kW m$^{-2}$ (see the orange line in Fig. 3g).

To further reveal the physical mechanism of evaporative cooling potential under various convection scenarios, we simulated the temperature reduction of the PAAS photonic film under various heat transfer coefficients and evaporation rates (Fig. 3h). Due to the high solar reflectance of PAAS photonic film, the simulated temperature reduction ($\Delta T = T_{film} - T_{ambient}$) retains prominence even when the non-radiative heat coefficient is above 20 W m$^{-2}$ K$^{-1}$ due to its high potential for evaporative cooling (the right-top part of Fig. 3h). The efficient solar scattering effect enables the PAAS photonic film to reflect sunlight, thereby reducing the heating effect. Additionally, its evaporative capability facilitates evaporative cooling, resulting in enhanced hybrid cooling performance. The unique combination of scattering and evaporation features introduces additional cooling channels besides radiative cooling. This results in a hybrid passive

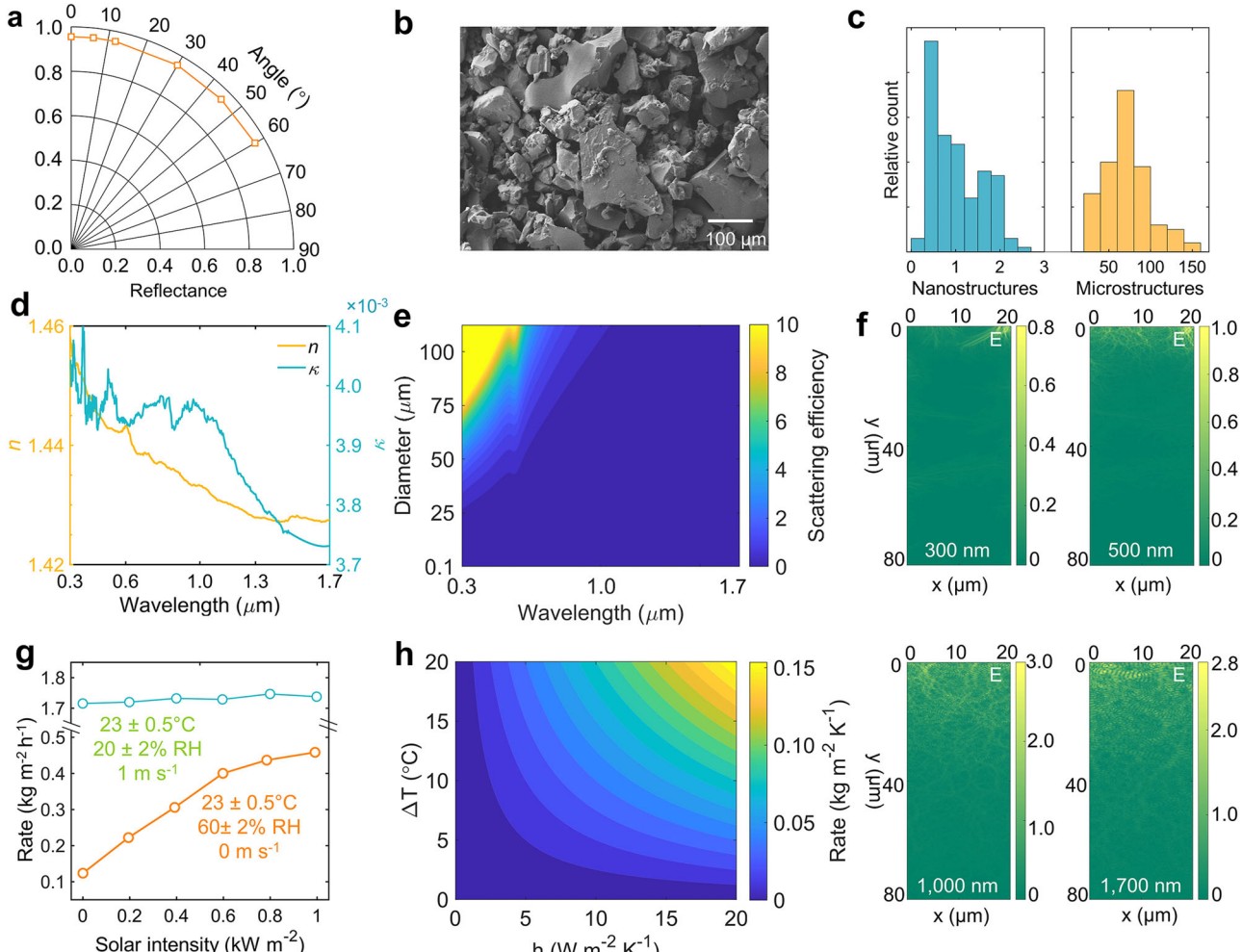

**Fig. 3 | Optical characterizations. a** Overall solar reflectance across different angles of incidence. **b** SEM images of the top orthography of the sodium polyacrylate (PAAS) photonic film. **c** Size distributions of the PAAS structures. **d** Complex refractive indices of the PAAS over the solar spectrum. **e** The scattering efficiency of the PAAS microstructures. **f** The electrical field of the PAAS photonic film in the dry states of four representative incident wavelengths, *E* represents the electricity field. **g** Evaporation rate of the PAAS photonic film as a function of solar intensity at different environment conditions (high humidity without wind and low humidity with the wind). **h** Simulated temperature reduction of different parasitic heat transfer coefficients when the evaporation rate of the PAAS photonic film varies.

cooling performance that is less dependent on weather conditions, as demonstrated in the following outdoor experiments.

## Radiative cooling and evaporative cooling performance demonstration

The spectral selectivity of PAAS photonic film over solar and thermal wavelengths equips itself with the potential to achieve hybrid subambient cooling (i.e., radiative cooling and evaporative cooling). To decouple the combined effect of these two cooling mechanisms, here, we separately measure the temperature reduction of PAAS photonic film by controlling the experimental settings.

For the radiative cooling measurement under a clear sky (Fig. 4a), we employ a polyethylene (PE) film over the experimental chamber at a location of 1 cm above the PAAS photonic film. The PE film is broadly transparent over both solar and infrared wavelengths (0.3–25 μm) and it does not block the sunlight and the infrared thermal dissipation (see the transmittance of PE film in Supplementary Note 6, Supplementary Fig. 23). This PE film also prevents the water absorption process of the dry PAAS photonic film. Therefore, the temperature drop is mainly introduced by the radiative cooling effect. As shown in Fig. 4b, we performed an outdoor experiment on Nov. 17, 2022, in Buffalo, New York with a solar intensity of approximately 400 W m⁻² from 3:00 PM

to 4:30 PM and an ambient temperature ranging from 16 °C to 20 °C during the experiment. The temperature reduction of the PAAS photonic film stabilizes around 3.7 °C below the ambient. Through the theoretical thermal balance analysis (see Supplementary Note 6), the net radiative cooling power is estimated to be 76 W m⁻² under this relatively cool weather condition (see the solid green curves in Fig. 4b).

To evaluate the evaporative cooling performance, we first monitor temperature reduction and evaporative cooling power in a temperature and humidity-controlled chamber (Fig. 4c). In this chamber, the PAAS sample adheres to a copper (Cu) plate (which acts as a heat spreader and supports the film), while a Kapton thin film heater is attached to the back of the Cu plate to provide heating power. The PAAS sample sides are covered with foam to exclusively characterize evaporative cooling from the top surface. As shown in Fig. 4d, a maximum temperature drop of approximately 6 °C is recorded 20 min after initiating the experiment, and the temperature stabilizes at 2.5 °C below ambient after roughly 130 min. The mass change continually increased, while its weight loss rate gradually decreased as evaporation continued from the top surface of the PAAS photonic film (Fig. 4e). This occurs because the time required for water from the bottom portion to diffuse to the top surface increases, decreasing the weight loss rate. As the water absorbed on the film's top surface evaporates, it

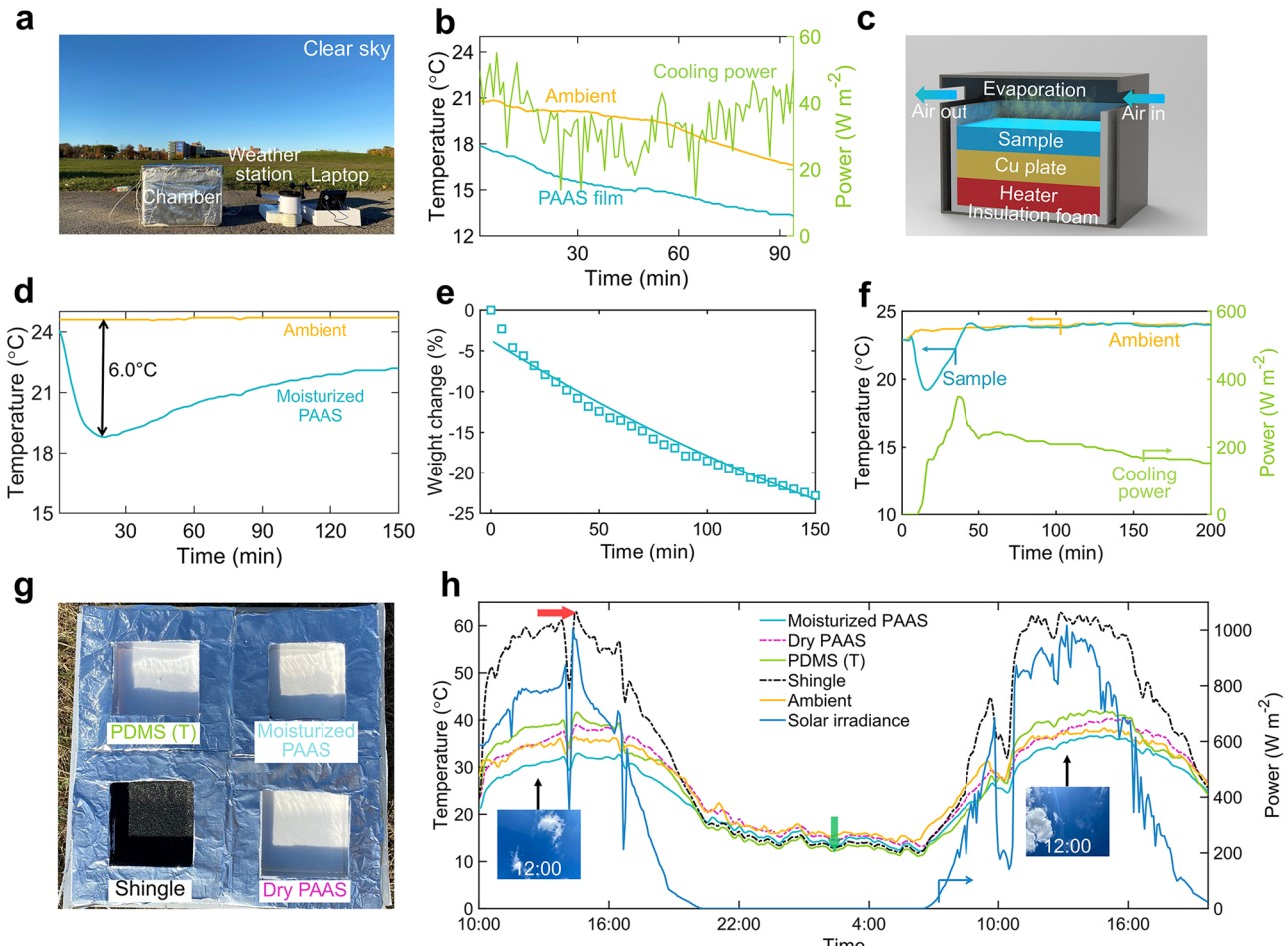

**Fig. 4 | Outdoor experiments. a** The outdoor radiative cooling experimental setup under direct sunlight. **b** Temperature variations of the sodium polyacrylate (PAAS) photonic film (only radiative cooling). The right y-axis plots the theoretical estimated radiative cooling power. **c** Schematic exhibiting the experimental setup of measuring the temperature drop and power of evaporative cooling. **d** Temperature reduction of the PAAS photonic film (only evaporative cooling). **e** Mass change of the PAAS photonic film during the evaporative cooling experiment. **f** Temperature and evaporative cooling power variations. **g** The setup of the outdoor hybrid cooling experiment. **h** Temperature variations of the roof shingle, PDMS (T), PAAS photonic film (wet), and PAAS (dry) film samples. Insets are photos of the sky in different weather conditions. The red arrow points out the maximum temperature of the shingle sample.

takes longer for water from the bottom part of the film to migrate to the top surface, extending the diffusion path to the top evaporative cooling surface. Consequently, the temperature reduction induced by evaporative cooling gradually diminishes. For the characterization of the evaporation cooling power, the sample is attached to a Cu plate on top of a heater, which is controlled by a PID controller to ensure that the PAAS sample surface temperature is in equilibrium with the ambient temperature (following the procedure reported ref. [31]). The heating power can be considered the cooling power because the back of the heater is thermally insulated. When the thin film heater and the PID controller are activated, the temperature of the PAAS sample is adjusted to match the ambient after 43 min (Fig. 4f, i.e., the blue curve overlaps the orange curve). Afterward, the evaporation cooling power fluctuates around 190 W m$^{-2}$ (green curve) and gradually decreases due to the water loss during continuous evaporation, which can be observed from the decreasing slope of the fitted curve in Fig. 4e.

Finally, we test dry and moisturized PAAS photonic films alongside two other sample-roof shingles, a common building material for roofs, and transparent PDMS coated on a polished aluminum plate (PDMS (T)), a radiative cooling material that was published in our previous work[30] under identical outdoor conditions to demonstrate the overall hybrid cooling performance of moisturized PAAS photonic film (Fig. 4g). The reflectance spectra of the four samples are presented in Supplementary Fig. 24. The thickness of the PAAS photonic film is

optimized to be 2 mm according to our preliminary experiment data (see Supplementary Fig. 25). This thickness holds the balance between enhanced mechanical robustness and increased moisture absorption, facilitating prolonged and continuous evaporative cooling.

During the experiment, the average solar intensity was 800 W m$^{-2}$ and the average relativity humidity was 60% (Supplementary Fig. 26). At noontime (indicated by the red arrow in Fig. 4h), the commercial shingle without PE cover film reaches 40 °C above ambient, further validating the need for photonic cooling materials to save energy. The dry PAAS sample reaches 2 °C above the ambient without evaporative cooling enhancement, while the PDMS (T) sample temperature is 4 °C above the ambient due to direct solar illumination (with a peak solar intensity of 920 W m$^{-2}$). In contrast, the PAAS photonic film exhibited a subambient temperature drop of 5 °C, demonstrating the superior performance of hybrid cooling in partly cloudy weather. During the nighttime (indicated by the green arrow in Fig. 4h), the PDMS (T) and shingle temperatures are both below that of the moisturized PAAS sample, even though they had similar infrared emittance spectra. Due to the moisture adsorption into the PAAS film at night, the regeneration process generated heat[51], resulting in a higher temperature compared to the PDMS (T) and shingle samples. As day and night alternated, the moisturized PAAS photonic film evaporates and absorbs moisture from the air, resulting in a hybrid cooling performance. The cooling performance of the moisturized PAAS photonic

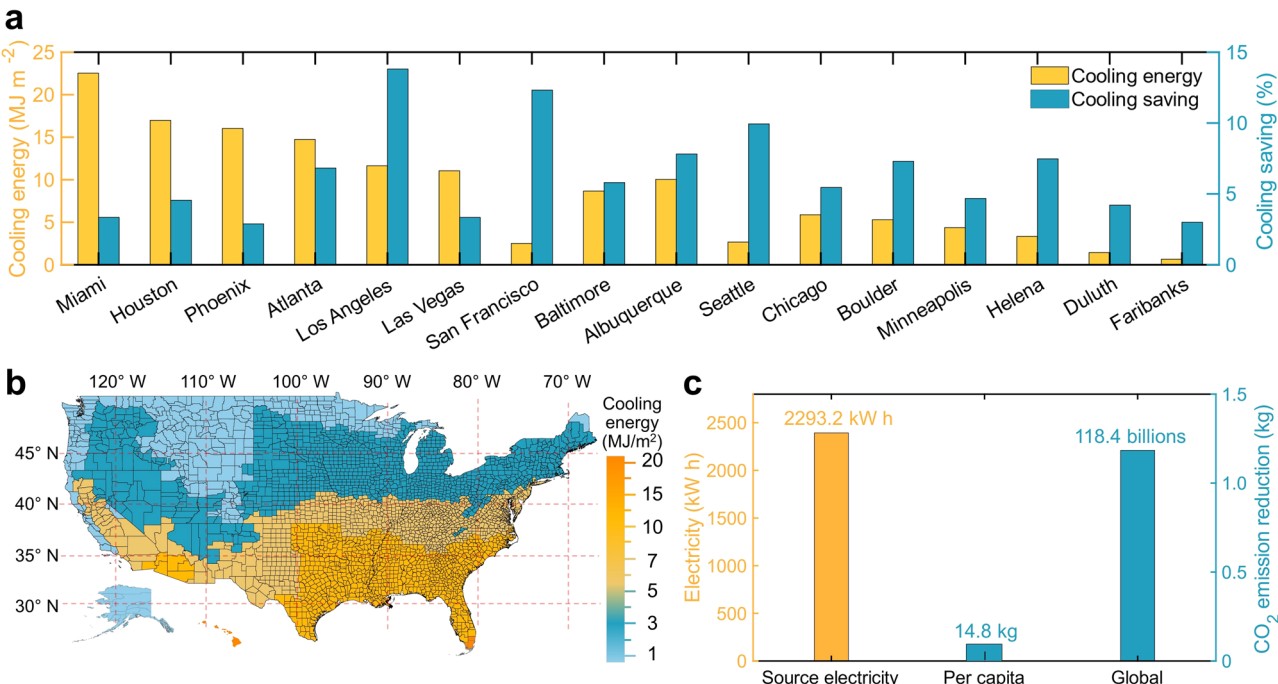

**Fig. 5 | Energy saving estimation. a** Estimated annual heat reduction per area for installation on walls and roofs for 16 representative cities in different climate zones. **b** Energy-savings map across the United States with the sodium polyacrylate (PAAS) cooling film. **c** Estimated average annual electricity and predicted average annual $CO_2$ emission reduction per capita and globally.

film surpassed that of dry PAAS and PDMS (T) with only radiative cooling capability, as well as commercial shingles with high solar absorption. Furthermore, under a clear sky, the temperature of the moisturized PAAS photonic film (with hybrid passive cooling) is 7.2 °C below the dry PAAS sample (with radiative cooling only), confirming the advantage of hybrid passive cooling over radiative cooling under a clear sky condition (Supplementary Fig. 27).

Furthermore, an additional 3-day experiment involving moisturized PAAS photonic film under diverse weather conditions—including a sunlit day with smoky air, a partially cloudy and warm day, and a predominantly cloudy day—serves to underscore the superior efficacy of the hybrid mechanism relative to pure radiative cooling (as illustrated in Supplementary Fig. 28). Notably, sustained cooling effectiveness assumes paramount importance in ensuring human comfort during daylight hours. With this in mind, we meticulously document the weight variations of the PAAS photonic film in the hybrid cooling mode. Our scrutiny reveals that the continuous water evaporation from the PAAS photonic film remains viable for an impressive span exceeding 9 h, as elucidated in Supplementary Fig. 29. This endurance reaffirms the capability of hybrid cooling to operate effectively throughout a significant portion of the daytime. It is imperative, however, to acknowledge the distinction between our experimental setup and practical scenarios. While our experiments lack the introduction of a consistent heat source, practical circumstances encompass continuous heat dissipation from electrical devices and human activities. Recognizing this discrepancy, we proceed to investigate the cooling performance of the PAAS photonic film under conditions of a steady heat source within the enclosure. As showcased in Supplementary Fig. 30, the results demonstrate the hybrid cooling of PAAS photonic film outperforms pure radiative cooling, elucidating the marked advantage of our enhanced heat dissipation capability compared to the conventional pure radiative cooling sample.

### The potential for energy-saving and carbon emission reduction

To assess the potential of the PAAS photonic film for sustainability, we developed an energy estimation model to quantify the energy savings

and carbon emission reductions that could be achieved by integrating the film onto the roof of a building (see supplementary notes S7 and Figs. S31–S33). Following a general building model defined by the US Department of Energy[52], we select the exterior roofs of a mid-rise apartment building and estimate its annual cooling reduction per area (see model details in Supplementary Note 7). As shown in Fig. 5a, we simulate the annual total reduction of heat gain and energy savings after installing PAAS photonic films on the roofs of 16 representative cities in various climate zones to examine its applicability. For instance, considering the long-term climate data in these selected cities (e.g., Miami), 22.5 MJ m$^{-2}$ of cooling energy can be saved per year, corresponding to 3.3% of the total energy consumption. More broadly, the simulated energy-saving map of over 100 representative US cities (with PAAS photonic film application on sky-facing roofs) is shown in Fig. 5b, using the same procedure while considering the long-term climate data[53]. One can see that up to 104.14 GJ of cooling energy (i.e., a mid-rise apartment of 3134.59 m$^2$ in Phoenix) can be saved for a typical midrise apartment building in a tropical climate zone when both the roof's exterior and walls are installed with PAAs photonic films, corresponding to 18.7% of the baseline energy used for cooling (see calculation details in Supplementary Note 7). Moreover, cooling energy-saving gains are more prominent in the southern parts of the United States (i.e., the yellow part is concentrated in the southern part of the U.S.), such as Miami and Los Angeles, compared to northern cities.

Furthermore, we calculate electricity and natural gas savings by converting the energy savings into equivalent electricity and natural gas based on the estimations from the EnergyPlus model simulation (see detailed settings in Supplementary Note 7). As a result, an average of 2293.2 kWh of electricity can be saved for a mid-rise apartment building (see the yellow bar in Fig. 5c), equaling an 887.9 kg $CO_2$ emission reduction and 14.8 kg per capita (i.e., the central blue bar). Based on the overall world population (i.e., the right blue bar), the approximate estimation of total annual global $CO_2$ emission reduction is 118.4 billion kg (following the estimation procedure used in ref. 54), representing a roughly 0.33% reduction from the current global $CO_2$ emissions (i.e., 36.4 billion tons per year[55]).

## Discussion

In summary, a spectrally selective film has been demonstrated for scalable hybrid passive cooling by employing atmospheric moisture-induced polyacrylate hydrogels. The moisture absorption facilitates the transfiguration of PAAS hydrogel from loose powders to continuous and flexible films. A dynamic water cycle is formed by moisture absorption at nighttime and evaporation cooling in the daytime. The strong hydrogel link that exists between neighboring PAAS particles guarantees mechanical robustness for long-term engineering applications. Furthermore, the dry PAAS photonic film exhibits a high reflectance of 0.93 over the broadband solar wavelengths resulting from the efficient backscattering of these randomized photonic pores. It also shows a high infrared thermal emittance of 0.99 over the atmospheric transparent window due to its intrinsic molecular vibrations. This spectral selectivity enables efficient passive cooling, featured by a temperature reduction of 5 °C under a partly cloudy sky observed at Buffalo, NY under a solar intensity of 800 W m$^{-2}$. This PAAS photonic film simultaneously brought evaporative cooling that is independent of access to the clear sky assisted by its hygroscopic feature. Compared to current electricity-driven air-conditioning facilities, this hybrid passive cooling strategy is expected to reduce global carbon emissions by 118.4 billion kg annually. The scalable and economical hygroscopic hydrogel has the potential to drive further development of passive cooling applications, including infrastructure cooling of electronic devices[56] and vehicles[57,58], and outdoor personal thermal management[59]. Moreover, oil tankers, refrigerated ships, and trucks that maintain relatively low temperatures to preserve food and temperature-sensitive goods could also benefit from our PAAS photonic film's passive cooling properties. Importantly, the entire atmospheric-moisture process utilizes no additional chemicals, making it a completely environmentally friendly and high-yield production with zero carbon emissions and even negative emissions from saved energy consumption in cooling.

## Methods

### Materials

Dry PAAS powder (99.995% pure, pharma grade) was provided by the DMSO store. PDMS (SYLGARD 184 silicone elastomer) is purchased from Dow. Thin-film heater, black Al ROSCO cinefoil, K-type thermocouples, weather station (WS 2000), temperature data logger (AZ 88598, 4 channels), and polystyrene thermal insulation foam were purchased commercially from Amazon.

### Materials characterizations

The surface and cross-section morphologies of the PAAS photonic film were taken using the SEM (Carl Zeiss Auriga FIB-SEM) with an acceleration voltage of 5 kV. An image processing software (ImageJ) was utilized to analyze those SEM images for the size distribution of PAAS film. The TGA curves were characterized from 20 °C to 300 °C with a temperature scanning rate of 5 °C min$^{-1}$ by Bruker TG 50. DSC analysis was carried out on a TA Instruments SDT Q600 from 18 °C to 350 °C under a heating rate of 5 °C min$^{-1}$. A thermal camera (FLIR C5) was used to take infrared images. An ellipsometer (J.A. Woolam, M200DI) was employed to extract the complex refractive indices of PAAS. The hardness of the PAAS sample was examined by the VTSYIQI Shore D durometer. The tensile strength was performed with the upper fixture moving downward at a constant velocity of 5 mm min$^{-1}$. The thermal conductivity of the PAAS film was characterized by Hot Disk TPS 2500 s.

### Optical characterizations

The hemispherical reflectance spectra of the PAAS photonic film were characterized by a spectrophotometer (Agilent Cary 7000) with a polytetrafluoroethylene-based integrating sphere (Model IS200, Thorlabs) over solar wavelengths (0.3–2.5 μm). The hemispherical reflectance spectra were measured by the FTIR spectrometer (Vertex 70) equipped with a golden integrating sphere (A562, Bruker) from 2.5 μm to 15 μm. The Raman spectra were examined by the BSTEK BWS 465 Raman spectrometer from 800 cm$^{-1}$ to 950 cm$^{-1}$. The film formation process was observed under a magnification ratio of 15 X utilizing Bruker Hyperion optical microscopy (IHYP 1126). The FTIR transmittance spectra were measured by the Jasco FTIR 6600 spectrometer from 4000 to 500 cm$^{-1}$.

### FDTD optical simulation

The electrical field simulation of PAAS photonic film was executed using the Lumerical FDTD Solution 2018a. The electrical and power field distributions of the PAAS film (20 × 80 μm) over the wavelength of 0.3–1.7 μm were simulated in a 2D FDTD model with a plane wave source. The randomized air-void structures were generated with dimensions of 20 × 80 μm in the x- and y-direction by converting the grayscale SEM images of the PAAS film into binary images using MATLAB R2020a (MathWorks, Inc.). The processed 2D images of randomized air-void structures were imported into the Lumerical FDTD solution as the refractive index profile. Perfectly matched layer (PML) boundary conditions were employed to absorb all the outgoing electromagnetic radiation. A plane wave source over wavelengths from 0.3 to 1.7 μm was launched at a distance of 5 μm away from the left edge of the HIRC metapaper simulation region and the refractive index of air voids was assigned as 1.

### Scattering efficiency calculation

The scattering efficiency of PAAS nanostructures, $\delta_{sca}$, is defined by

$$\delta_{sca} = \frac{8\pi}{3}\kappa^4 r^4 \left| \frac{\epsilon - \epsilon_m}{\epsilon + 2\epsilon_m} \right|^2$$

where $\kappa = 2\pi/\lambda$ represents the wave vector, and $\lambda$ means the wavelength. $r$ defines the radius of the nanoparticles. $\epsilon$ is the complex dielectric function of the PAAS and $\epsilon_m$ is the dielectric constant of the surrounding medium (air).

## Data availability

Source data are provided with this paper.

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

## Acknowledgements

This work was partially supported by a baseline from the King Abdullah University of Science and Technology (BAS/1/1415-01) and the Competitive Research Grant (URF/1/5019-01-01).

## Author contributions

Q.G. conceived the idea and supervised this project. R.G. prepared the samples, materials characterizations, and thermal-related experiments. Y.T. conducted the optical simulations, Y. L. built the Energy-saving modeling, and X.L. and A.Y. carried out the mechanical tests. S.D. and G.L. performed mechanical characterizations and discussed the corresponding results. All authors contributed to the analysis of the experimental results and modeling. Y.T. and Q.G. wrote the manuscript. All authors reviewed and revised the manuscript.

## Competing interests

A disclosure with Q.G. as an inventor is under consideration and processing by the KAUST Technology Transfer Office (Application/Provisional Number: US 63/524,948). All other authors have no competing interest.
