## [Peer Review File · Nature Communications]

REVIEWER COMMENTS

Reviewer #1 (Remarks to the Author):

This manuscript deals with hybrid cooling based on radiation and evaporation by a photonic hydrogel, which is generated by transforming sodium polyacrylate (PAAS) powder into a white film using atmospheric moisture. The proposed PAAS photonic film show strong sunlight reflection as well as high emissivity in the mid-infrared ranges, thereby providing superior cooling performance compared to that of conventional passive radiative cooling materials. The films also have self-healing properties, which is helpful for long-term usability. All the manufacturing process support net negative carbon emissions. Overall, it demonstrated a possibility that can help an effective cooling in humid weather or in urban areas, where the conventional radiative cooling makes failures. Thus, the reviewer recommends acceptance if comments below can be properly addressed.

- Could you provide a detailed comparison between a PAAS powder and others in terms of optical/thermal properties? The reviewer is wondering what the alternative materials are.
- For the practical use of the proposed materials, is it possible to add colors on that? Several researchers are trying to develop a colored passive radiative cooler (CPRC).
- The authors mentioned that the proposed materials could be used in urban areas, but didn't provide specific applications. There have been many reports on the radiative cooling in various applications, including buildings, cars, wearable devices, solar cells, and energy harvesters. It would be great if the authors could comment on the potential applications in the conclusion section.
- There have been several articles on the heat release from the enclosures based on radiative cooling. I believe the combination with evaporation could accelerate the heat extraction. There would have quite different cooling behaviors with that of Figure 4. The authors could comment on that.
- Some words in figures are too small.

Reviewer #2 (Remarks to the Author):

The article presents a new material that can provide cooling through radiative and evaporative processes. The article is innovative and merits to be considered for publication provided the authors improve its quality and provide the following information / explanations :

1. How the optical characteristics, reflectance and emissivity, change as a function of the absorbed water vapor?
2. For how long during the day, the material keeps its superior characteristics given the evaporation of the water vapor?
3. The experimental results are quite poor and refer just to some hours of testing. The material has to be tested for much longer periods and under various climatic and boundary conditions.
4. Which is the required threshold of the night time ambient humidity ?
5. It is claimed that the material helps to overcome the performance limitations of radiative coolers in humid climates. How ?
6. Evaporation during the daytime may affect the performance of the radiative cooler especially if the humidity around the cooler is quite high.

Reviewer #3 (Remarks to the Author):

This manuscript presents a comprehensive study on a hydrogel designed for atmospheric-moisture-induced evaporative and radiative cooling. The authors provide a robust background and detailed results, introducing a photonic PAAS that exhibits not only exceptional optical properties but also reusability and self-healing capabilities. I recommend the publication of this manuscript in Nature Communications, subject to minor revisions.

1. In Figure 3e, the term 'wavelength' appears to be obscured by the figure below. Could the authors please rectify this?
2. The manuscript suggests that PAAS microstructures exhibit a scattering efficiency independent of diameter. This result seems counterintuitive. Could the authors please verify this and provide further clarification?
3. Figure 4b illustrates the cooling power and temperatures of ambient air and the PAAS film. To measure cooling power, heat must be applied to the PAAS film until it reaches ambient air temperature. Could the authors elaborate on the methodology used to measure this?
4. The PAAS film appears to have a thickness exceeding 1 mm, which is considerably thicker than many radiative cooling films. The thickness of radiative cooling films can have both advantages and disadvantages. Could the authors discuss the impact of the thickness of their PAAS film on its performance?

RESPONSE TO REVIEWERS' COMMENTS

Reviewer #1:

This manuscript deals with hybrid cooling based on radiation and evaporation by a photonic hydrogel, which is generated by transforming sodium polyacrylate (PAAS) powder into a white film using atmospheric moisture. The proposed PAAS photonic film shows strong sunlight reflection as well as high emissivity in the mid-infrared ranges, thereby providing superior cooling performance compared to that of conventional passive radiative cooling materials. The films also have self-healing properties, which is helpful for long-term usability. All the manufacturing processes support net negative carbon emissions. Overall, it demonstrated a possibility that can help an effective cooling in humid weather or in urban areas, where conventional radiative cooling makes failures. Thus, the reviewer recommends acceptance if the comments below can be properly addressed.

[Response]: The authors sincerely express their gratitude to the reviewer for dedicating their valuable time to review our manuscript and providing valuable feedback to affirm and guide our work. Kindly find our point-by-point response listed below:

1. Could you provide a detailed comparison between a PAAS powder and others in terms of optical/thermal properties? The reviewer is wondering what the alternative materials are.

[Response]: In this revision, we have included a detailed comparison between PAAS photonic film and other similar materials in terms of their optothermal properties, as described in **Table S1** (see page 5 of the supplementary information and lines 63-66 on page 2 of the main text). Polymeric hydrogels, such as PAAS, pNIPAm, PVA, and polyacrylamide (PAM), are hygroscopic substances that can be employed as matrix materials, while LiCl, LiBr, or CaCl₂ can be used to enhance the absorption of moisture in the air.

Table S1. Comparison of optical/thermal properties of PAAS with other literature. The values marked with an asterisk (*) indicate estimates based on the information provided in the corresponding reference.

Reference	Solar reflectance	Thermal emissivity	Structure	Fabrication method	Composition
Sodium polyacrylate (PAAS) ^{This work}	0.93	0.99	Single layer	Atmospheric moisture absorption	100% PAAS
Hierarchical Porous Coating (HPC) ^[R1]	0.95	0.98	Single layer	solvent exchange method	Modified Poly(vinylidene fluoride-co-hexafluoropropene) (PVDF) and Cellulose Acetate (CA) polymeric network
poly(N-isopropylacrylamide) (pNIPAm) ^{[R2]*}	0.92	0.94	Three-part sandwich structure thermal homeostasis (SSTH)	free radical UV-initiated polymerization technique	pNIPAM sandwich

poly (vinyl alcohol) (PVA)–CaCl ₂ [R3]	0.94	0.94	bilayer polymer	freeze-drying/electrostatic-spinning synthesis method	cellulose acetate (CA) network and poly(vinyl alcohol) (PVA)–CaCl ₂ hydrogel
LiBr-polyacrylamide (PAAm) – poly(vinylidene fluoride-co-hexafluoropropene) [P(VdF-HFP)] [R4]	0.96	0.96	bilayer porous polymer	Photopolymerization and phase inversion method	Li-PAAm hygroscopic hydrogel and poly(vinylidene fluoride-co-hexafluoropropene) porous hydrophobic layer

Reference

R1. Fei, J., Han, D., Ge, J., Wang, X., Koh, S. W., Gao, S., ... & Li, H. (2022). Switchable surface coating for bifunctional passive radiative cooling and solar heating. *Advanced Functional Materials*, 32(27), 2203582.

R2. Fang, Zhen, Liyun Ding, Lintao Li, Kun Shuai, Boyu Cao, Yetao Zhong, Zhenghua Meng, and Zhilin Xia. "Thermal homeostasis enabled by dynamically regulating the passive radiative cooling and solar heating based on a thermochromic hydrogel." *ACS Photonics* 8, no. 9 (2021): 2781-2790.

R3. Li J, Wang X, Liang D, Xu N, Zhu B, Li W, Yao P, Jiang Y, Min X, Huang Z, Zhu S, Fan S, Zhu J. A tandem radiative/evaporative cooler for weather-insensitive and high-performance daytime passive cooling. *Sci Adv.* 2022 Aug 12;8(32)

R4. Feng, Chunzao, Peihua Yang, Huidong Liu, Mingran Mao, Yipu Liu, Tong Xue, Jia Fu et al. "Bilayer porous polymer for efficient passive building cooling." *Nano Energy* 85 (2021): 105971.

2. For the practical use of the proposed materials, is it possible to add colors to them? Several researchers are trying to develop a colored passive radiative cooler (CPRC).

[Response]: We appreciate the suggestion regarding the practical use of PAAS photonic film and the possibility of enabling them to be colored for aesthetic purposes.

Indeed, several researchers have demonstrated potential strategies for integrating aesthetic functions into passive radiative cooling applications [R5-R10], which further broadens the deployment of passive cooling techniques. Three prominent methods are proposed for introducing various colors to radiative cooling materials, such as embedding fluorescent or dye nanoparticles into materials during or after fabrication of the materials [R8, R10, R11] and constructing dielectric resonance photonic structures [R5] or metamaterials [R9] for generating structural color. However, several challenges need to be addressed for further development of colorful passive radiative materials. For instance, broadband visible absorption peaks of the resonant structures will make the colors less bright, and the broadband visible wavelength absorption will significantly increase the solar heating effect [R12]. Moreover, structure colors from multilayer metamaterials require cost- and labor-ineffective vacuum deposition techniques which remarkably hinder its scale-up implementations [R11]. Structurally colored film prepared from naturally derived cellulose

nanocrystals is also demonstrated for only colored passive radiative cooling [R10]. These colorful radiative cooling techniques only focus on the radiative cooling area. For now, no colorful passive radiative and evaporative cooling approaches have been reported to the best knowledge of the authors.

To validate the versatility of our proposed atmospheric-moisture-induced fabrication method, we introduce color pigments (Rolio© mica powder) by mixing them with PAAS power powder (1:100 ratio) and then putting them in an environment-controlled chamber (RH, ~ 90%) to form a continuous film with different colors. This procedure is similar to the one described in our manuscript and the only difference is the introduction of pigment powders. The introduced pigment powders did not affect the formation of continuous film, as depicted in **Fig. R1a** where four colored PAAS photonic films with different colors are displayed. These films show bright colors to the naked eye which corresponds to the absorption peaks as shown in the spectra results in **Fig. R1b**. Moreover, compared with the pristine white PAAS photonic film, these colored PAAS films keep high reflectance over other solar wavelengths (e.g., near-infrared) while displaying unity thermal emittance over infrared wavelengths. The high infrared thermal emittance enables effective radiative heat dissipation through the atmospheric transparent window. This moisture-induced technique for colored photonic film allows for more flexibility in terms of achieving specific color effects or incorporating multiple colors.

Figure R1. (a) Photos displaying the continuous PAAS photonic film of four different colors (red, yellow, green, and blue). (b) Spectra of the colored PAAS photonic film to demonstrate its aesthetic functionality while maintaining effective radiative cooling capabilities.

Furthermore, a colorful appearance will be more attractive for specific applications, such as wearable electronics, automotive and cooling textiles. However, adding pigment powders to the white radiative cooler may compromise the high solar reflectivity and hence weaken the cooling performance of the film. Therefore, other approaches that can introduce narrowband visible wavelength absorption are highly demanded by further hybrid passive cooling technologies.

In this revision, we added this potential in lines 131-134 on page 4 and included the results on page 10 of the supplementary information.

Reference:

R5. Lee, G. J., Kim, Y. J., Kim, H. M., Yoo, Y. J., & Song, Y. M. (2018). Colored, daytime radiative coolers with thin-film resonators for aesthetic purposes. *Advanced Optical Materials*, 6(22), 1800707.

R6. Photonic thermal management of coloured objects, *Nature Communications* volume 9, Article number: 4240 (2018)

R7. Colored and paintable bilayer coatings with high solar-infrared reflectance for efficient cooling, DOI: 10.1126/sciadv.aaz5413

R8. Colored radiative cooling coatings using phosphor dyes, <https://doi.org/10.1016/j.mtnano.2022.100239>

R9. Colored passive daytime radiative cooling coatings based on dielectric and plasmonic spheres, <https://doi.org/10.1016/j.applthermaleng.2022.119125>

R10. Highly efficient subambient all-day passive radiative cooling textiles with optically responsive MgO embedded in porous cellulose acetate polymer , <https://doi.org/10.1016/j.cej.2023.143765>

R11. Colorful surfaces for radiative cooling Lyu Zhou, Jacob Rada, Haomin Song, Boon Ooi, Zongfu Yu, and Qiaoqiang Gan

R12. Colored radiative cooling: progress and prospects, <https://doi.org/10.1016/j.mtener.2023.101302> Wang, Tong, Yinan Zhang, Min Chen, Min Gu, and Limin Wu. "Scalable and waterborne titanium-dioxide-free thermochromic coatings for self-adaptive passive radiative cooling and heating." *Cell Reports Physical Science* 3, no. 3 (2022).

3. The authors mentioned that the proposed materials could be used in urban areas but didn't provide specific applications. There have been many reports on radiative cooling in various applications, including buildings, cars, wearable devices, solar cells, and energy harvesters. It would be great if the authors could comment on the potential applications in the conclusion section.

[Response]: We appreciate this constructive comment. It is noted that radiative cooling has demonstrated significant potential in a wide range of applications, including buildings, automobiles, wearable devices, solar cells, and energy harvesters.

In this revision, we have thoroughly assessed the potential applications of our proposed material in the conclusion section. While wearable electronics and personal thermal management fabric could benefit from radiative cooling, we have included these applications after the demonstration of the versatility of the colored PAAS photonic film (see lines 131-134 on page 4). In addition, we envision that our proposed PAAS photonic film can be employed on large surfaces directly exposed to the sky, such as building envelopes [R13-R17], electrical cabinets [R18-R25], camping tents [R26-R28], and vehicles [R29-R32]. One significant advantage is that our material can be directly integrated into existing surfaces without extensive installation procedures because of the stickiness of PAAS hydrogel film.

Building envelopes, in particular, represent a straightforward and impactful application scenario for our PAAS photonic film, as it has the potential to reduce a substantial amount of energy consumption and CO₂ emissions, as demonstrated in Figure 5 of our manuscript. However, we acknowledge that the cooling performance of our material will depend strongly on location and weather conditions. It is expected to excel in hot regions such as the Middle East, China, and the southern parts of the US, while its performance may not be as ideal in Europe, Canada, and the northern parts of the US [R28-R29].

Electrical vehicles, which are known to have reduced mileage due to air conditioning, represent another potential market for our PAAS photonic film. It is particularly attractive for oil tankers, refrigerated ships, and trucks where maintaining relatively low temperatures is crucial [R30-R31]. Additionally, storage containers, especially those requiring a cool environment for preserving food and temperature-sensitive goods, could greatly benefit from our material's cooling properties. Furthermore, our PAAS material holds

promise for applications in cooling solar cells and LEDs, as these devices often require substantial energy to regulate their temperature around the optimal working temperature [R32].

In this revision, we added the discussion of potential applications in the conclusion section (i.e., lines 478-482 on page 17) and cited representative applications from ref. [57] to [60] in the main text.

Reference:

R13. Zhai, Y., Ma, Y., David, S. N., Zhao, D., Lou, R., Tan, G., ... & Yin, X. (2017). Scalable-manufactured randomized glass-polymer hybrid metamaterial for daytime radiative cooling. *Science*, 355(6329), 1062-1066.

R14. Raman, A. P., Anoma, M. A., Zhu, L., Rephaeli, E., & Fan, S. (2014). Passive radiative cooling below ambient air temperature under direct sunlight. *Nature*, 515(7528), 540-544.

R15. Zhou, L., Song, H., Liang, J., Singer, M., Zhou, M., Stegenburgs, E., ... & Gan, Q. (2019). A polydimethylsiloxane-coated metal structure for all-day radiative cooling. *Nature Sustainability*, 2(8), 718-724.

R16. Hong, S., Gu, Y., Seo, J. K., Wang, J., Liu, P., Meng, Y. S., ... & Chen, R. (2019). Wearable thermoelectrics for personalized thermoregulation. *Science advances*, 5(5), eaaw0536.

R17. Wang, J., Tan, G., Yang, R., & Zhao, D. (2022). Materials, structures, and devices for dynamic radiative cooling. *Cell Reports Physical Science*.

R18. Zhu, L., Raman, A., Wang, K. X., Abou Anoma, M., & Fan, S. (2014). Radiative cooling of solar cells. *Optica*, 1(1), 32-38.

R19. Wang, K., Luo, G., Guo, X., Li, S., Liu, Z., & Yang, C. (2021). Radiative cooling of commercial silicon solar cells using a pyramid-textured PDMS film. *Solar Energy*, 225, 245-251.

R20. Fan, S., & Li, W. (2022). Photonics and thermodynamics concepts in radiative cooling. *Nature Photonics*, 16(3), 182-190.

R21. Ishii, S., Dao, T. D., & Nagao, T. (2020). Radiative cooling for continuous thermoelectric power generation in day and night. *Applied physics letters*, 117(1).

R22. Zhao, B., Pei, G., & Raman, A. P. (2020). Modeling and optimization of radiative cooling based thermoelectric generators. *Applied Physics Letters*, 117(16).

R23. Liu, J., Zhang, Y., Zhang, D., Jiao, S., Zhang, Z., & Zhou, Z. (2020). Model development and performance evaluation of thermoelectric generator with radiative cooling heat sink. *Energy Conversion and Management*, 216, 112923.

R24. Liu, Y., Hou, S., Wang, X., Yin, L., Wu, Z., Wang, X., ... & Cao, F. (2022). Passive radiative cooling enables improved performance in wearable thermoelectric generators. *Small*, 18(10), 2106875.

R25. Kang, M. H., Lee, G. J., Lee, J. H., Kim, M. S., Yan, Z., Jeong, J. W., ... & Song, Y. M. (2021). Outdoor - useable, wireless/battery - free patch - type tissue oximeter with radiative cooling. *Advanced Science*, 8(10), 2004885.

- R26. Pu, S., Fu, J., Liao, Y., Ge, L., Zhou, Y., Zhang, S., ... & Chen, J. (2020). Promoting energy efficiency via a self - adaptive evaporative cooling hydrogel. *Advanced Materials*, 32(17), 1907307.
- R27. Li, Z., Ma, T., Ji, F., Shan, H., Dai, Y., & Wang, R. (2023). A Hygroscopic Composite Backplate Enabling Passive Cooling of Photovoltaic Panels. *ACS Energy Letters*, 8(4), 1921-1928.
- R28. Peng, Y., Chen, J., Song, A. Y., Catrysse, P. B., Hsu, P. C., Cai, L., ... & Cui, Y. (2018). Nanoporous polyethylene microfibrils for large-scale radiative cooling fabric. *Nature sustainability*, 1(2), 105-112.
- R29. Sui, C., Pu, J., Chen, T. H., Liang, J., Lai, Y. T., Rao, Y., ... & Hsu, P. C. (2023). Dynamic electrochromism for all-season radiative thermoregulation. *Nature Sustainability*, 6(4), 428-437.
- R30. Lv, Y., Huang, A., Yang, J., Xu, J., & Yang, R. (2021). Improving cabin thermal environment of parked vehicles under direct sunlight using a daytime radiative cooling cover. *Applied Thermal Engineering*, 190, 116776.
- R31. Mousavi, N. S., & Azzopardi, B. (2023). Perspectives on the Applications of Radiative Cooling in Buildings and Electric Cars. *Energies*, 16(14), 5256.
- R32. Hsiao, T. J., Eyassu, T., Henderson, K., Kim, T., & Lin, C. T. (2013). Monolayer graphene dispersion and radiative cooling for high power LED. *Nanotechnology*, 24(39), 395401.

4. There have been several articles on the heat release from the enclosures based on radiative cooling. I believe the combination of evaporation could accelerate heat extraction. There would have quite different cooling behaviors from that of Figure 4. The authors could comment on that.

[Response]: We appreciate the insightful commentary concerning the potential differentials in cooling behavior that emerge through the amalgamation of evaporation and radiative cooling. We agree with the reviewer that the combination of these cooling mechanisms can lead to distinct cooling effects in enclosures compared to the results shown in Figure 4. It is essential to recognize the dichotomy between heat transfer in open spaces, predominantly orchestrated by conduction and convection, and the nuanced mechanisms at play within enclosed environments where heat tends to accumulate. This is especially pertinent in scenarios encompassing outdoor electronics and parked vehicles.

To empirically validate this proposition, we conducted temperature recording experiments covering an aluminum enclosure with PAAS photonic film under clear sky conditions (**Fig. R2a**). The experimental setup encompassed three hollow enclosures, each fitted with a distinct cover: (1) a bare enclosure, (2) an enclosure clads with radiative PAAS film, and (3) an enclosure swathed in a hybrid PAAS film. The experimental protocol involved the application of a constant heat flux generated by a DC power supply. This power supply, calibrated to operate at a voltage of 3.0 V and a current of 0.25 A, effectively rendered a consistent heat flux of 300 W m^{-2} , simulating the thermal output of electronic devices. Noteworthy is the consistent and unwavering nature of this heating power throughout the experimental duration.

The outcomes unveiled that the temperature of the bare enclosure surpassed ambient temperatures by 9°C . Conversely, the application of PAAS films – both radiative and hybrid – heralded commendable sub-ambient performance. Most strikingly, the hybrid PAAS film showcased a substantial temperature mitigation of 6.1°C compared to the bare enclosure (**Fig. R2b**). This salient observation underscores the efficacy of the PAAS film in dissipating entrapped heat and in thwarting the ingress of solar radiation into the enclosure. Evidently, the convergence of evaporation and radiative cooling, epitomized by our hybrid

PAAS film, manifests as a distinctive and heightened cooling demeanor, an attribute with manifold advantages across diverse applications (especially in warm and humid areas).

In this revision, we have added the related experimental results on page 38 of supplementary information and discussed them in lines 421 to 428 on page 15 of the main text.

Figure R2: (a) Schematics showing the experimental setup. (b) Temperature response of the bar Al enclosure, and with radiative cooling or hybrid cooling PAAS photonic film.

5. Some words in figures are too small.

[Response]: We have revised all figures in the main text and increased the text size to ensure better readability. Please refer to the revised manuscript.

Reviewer #2:

The article presents a new material that can provide cooling through radiative and evaporative processes. The article is innovative and merits to be considered for publication provided the authors improve its quality and provide the following information/explanations:

[Response]: The authors sincerely express their gratitude to the reviewer for dedicating invaluable time and effort to meticulously review our manuscript. Additionally, we deeply appreciate the invaluable suggestions provided. For a comprehensive and detailed response to the reviewer's comments, kindly refer to our point-by-point response provided below:

1. How do the optical characteristics, reflectance, and emissivity, change as a function of the absorbed water vapor?

[Response]: Presented in **Figure R3** are the changes in optical properties, including reflectance and emissivity, in relation to water vapor. As the water content in the PAAS photonic film increases, a noteworthy effect is observed: the difference in refractive index between PAAS and ambient air diminishes, consequently weakening the backscattering of sunlight. This phenomenon leads to a reduction in reflectance across visible wavelengths. Remarkably, the moisturization of the PAAS photonic film results in a discernible decrease in its scattering effect. It is worth noting that variations in water content do not induce distinguishable differences over infrared wavelengths, as evident in the thermal emittance spectra depicted in Figure R3. These spectra illustrate a remarkable overlap among the curves, indicating negligible changes in the film's behavior in the infrared region for different water contents.

In this revision, we included the optical responses of PAAS with different water contents on page 25 of the supplementary information and discussed this on lines 276 to 279 on page 9 of the main text.

Figure R3: Spectra of the PAAS photonic film with different water contents.

2. For how long during the day, the material keeps its superior characteristics given the evaporation of the water vapor?

[Response]: The material's effective working duration is notably influenced by prevailing weather conditions. In order to demonstrate its exceptional cooling ability, we conducted a rigorous experiment on July 12, 2023, specifically focusing on measuring the temperature reduction achieved solely through evaporative cooling. This experiment was carried out in Buffalo, NY. During the experiment, we observed that the PAAS photonic film exhibited remarkable evaporative cooling performance for a minimum of 9 hours and possibly even longer. To accurately gauge the cooling effect, we devised an experimental setup

closely resembling the outdoor test described in our manuscript. The only deviation was the use of an aluminum cover, which effectively blocked access to the sky and thereby prevented radiative cooling. However, to facilitate the escape of vapor for evaporative cooling, we left a gap between the aluminum cover and the sample.

The temperature response of the PAAS photonic film was carefully recorded and is presented in **Fig. R4a**. Notably, throughout the duration of the experiment, the film's temperature consistently remained below the ambient temperature, unequivocally validating the occurrence of evaporative cooling from approximately 11 am to 9 pm. Additionally, we monitored the water content of the PAAS photonic film during the experiment. As depicted in **Fig. R4b**, the water content steadily decreased from 0.7 g/g to nearly zero, demonstrating its consistent ability to sustain water evaporation throughout the day. This reliable evaporation process effectively maintained a temperature reduction from the ambient during the daytime, when the demand for cooling is more significant than at nighttime. For a comprehensive understanding of the experimental conditions, we also illustrated the variations in relative humidity and solar intensity over the entire experimental period in **Fig. R4c**. These details provide essential context and further support the validity of our findings.

In this revision, we have included the performance of water evaporation of PAAS photonic film on lines 415 to 421 in the main text and added them on page 37 of the supplementary information.

Figure R4: (a) Temperature response of the PAAS photonic film when only evaporative cooling happens. (b) Water content changes over the daytime of the PAAS photonic film. (c) Relative humidity and solar intensity variation over the experimental period.

3. The experimental results are quite poor and refer just to some hours of testing. The material has to be tested for much longer periods and under various climatic and boundary conditions.

[Response]: We appreciate this insightful comment. In this revision, the authors performed an extra **three-day experiment** and provided the temperature variations of dry PAAS, PDMS (T), Shingle, and moisturized PAAS, as described in Figure 4g of the main text. At noontime (indicated by the red arrow in Figure 4h), the commercial shingle without PE cover film reaches 40°C above ambient, further validating the need for photonic cooling materials to save energy. The dry PAAS sample reaches 2°C above the ambient without evaporative cooling enhancement, while the PDMS (T) sample temperature is 4°C above the ambient due to direct solar illumination (with a peak solar intensity of 920 W m⁻²). In contrast, the PAAS photonic film exhibited a subambient temperature drop of 5°C, demonstrating the superior performance of hybrid cooling in partly cloudy weather.

To further demonstrate the superior cooling performance of PAAS photonic film, we tested our materials for longer periods of time under various climatic and boundary conditions. Specifically, we performed uninterrupted tests for three consecutive days with different weather conditions, as shown in **Fig. R5**.

On Day 1, we tested our material in a sunny sky with smoky air, due to suspended pollutant particles (PM2.5 average particle size 2.5 micron) from wildfires, which led to unhealthy air quality (AQI 150+) and scattering of PAAS photonic film emissions to the sky. Consequently, the radiative cooling effect was hindered, and the radiative cooler only reached around ambient temperature. However, the hybrid cooler exhibited a mean 4°C subambient cooling at noontime, due to its evaporative cooling functionality, which remained insensitive to this extreme weather condition (see day 1 result in **Fig. R5a**).

On Day 2, we tested the material under a partly cloudy sky and hot day. The mean solar intensity was ~ 850 W m⁻² (see day 2 result in **Fig. R5b**) and the average ambient temperature was 32°C. Even under such conditions, the hybrid cooling material realized a subambient cooling of 8°C under peak solar intensity, outperforming the radiative cooling material by 4°C.

Lastly, on Day 3, we measured the cooling performance on a mostly cloudy where radiative heat transfer is blocked to the universe. As expected, the radiative cooler struggled to achieve sub-ambient cooling, reaching the ambient temperature at noontime. However, the hybrid cooler still maintained a temperature ~5°C below ambient, highlighting its capability to perform well under different extreme weather conditions. Our results show the hybrid cooler performs well under different weather conditions (see day 3 result in **Fig. R5b**).

In this revision, we added these new data on pages 35-36 of supplementary information and discussed this on lines 412 to 415 in the main text on pages 14 and 15.

Figure R5: (a) Temperature variation of the PAAS photonic film under two scenarios: hybrid cooling and radiative cooling. (b) The corresponding solar intensity and humidity during the 3-day experimental period.

4. Which is the required threshold of the nighttime ambient humidity?

[Response]: Under our test environment with the ambient temperature of 22°C, the successful fabrication and regeneration of the PAAS film necessitate a minimum nighttime ambient humidity threshold of 60% RH or higher. In order to establish this pivotal threshold, in this revision, we undertook an experimental study involving a PAAS powder sample. This sample was positioned within a controlled environmental chamber, which was calibrated to varying relative humidity levels: 30%, 50%, 55%, 60%, 70%, and 90%. These conditions were maintained at an ambient temperature of 22°C for a duration of 6 hours. Following the 6-hour interval, a methodical assessment was conducted by delicately peeling off the PAAS film from the substrate. This examination aimed to gauge both the mechanical stability and the continuity of the film. The outcomes of this investigation are depicted in **Fig. R6**. Notably, at 30% RH, no film formation was observed; the PAAS remained in its powdered state. At 50%~55% RH, some degree of PAAS film formation was evident, albeit with discontinuous films showcasing inadequate mechanical stability. However, a significant transition was observed from 60% RH onward. At this threshold and beyond, a consistent and continuous film formation emerged, characterized by commendable mechanical stability. In light of these findings, we arrived at a conclusive determination: the requisite threshold for achieving successful PAAS film fabrication and regeneration stands at 60% RH or higher. In this revision, we **discussed the minimum relative humidity to form a continuous film in lines 128 to 130 on page 4 of the main text and included the experimental details on page 9 of the supplementary information.**

Figure R6: Photos showing the continuous film formation of PAAS powders at different relative humidity.

5. It is claimed that the material helps to overcome the performance limitations of radiative coolers in humid climates. How?

[Response]: It is known that in climates with high water vapor levels, the atmospheric transmittance, which serves as a heat transfer bridge between Earth's surface objects and the cold outer space, is significantly reduced. This reduction in atmospheric transmittance can undermine the pure radiative cooling potential. Consequently, it becomes imperative to adopt alternative strategies to reinforce passive cooling mechanisms. One such effective approach is evaporative cooling, a process wherein the temperature of a substance or object is reduced through the evaporation of water or other liquids. During evaporation, heat is absorbed from the surroundings, leading to a cooling effect. In this study, we employ PAAS (Polyacrylic acid sodium salt), a desiccant capable of absorbing moisture from the air during the nighttime and releasing it through evaporation during hot days. This unique property enhances the radiative cooling force and helps mitigate the impact of higher relative humidity on pure radiative cooling. Furthermore, the use of PAAS powder proves advantageous due to its proficiency in backscattering sunlight, which effectively reduces

the solar heating effect. Additionally, the material exhibits a high thermal emittance over the atmospheric window, thus accelerating heat dissipation. These combined attributes make PAAS a promising candidate for bolstering passive cooling capabilities and promoting energy-efficient cooling solutions. **In this revision, we clarified this potential in lines 86-88 on page 3 of the main text.**

6. Evaporation during the daytime may affect the performance of the radiative cooler especially if the humidity around the cooler is quite high.

[Response]: We greatly appreciate the valuable feedback from the reviewer and have taken into account their concerns regarding the performance limitations of our hybrid cooler in humid climates. We agree with the reviewer that evaporation during the daytime may increase the relative humidity over the sample and then affects the radiative cooling component of the hybrid cooler, particularly when ambient relative humidity is high.

To validate that the RH increase over the PAAS film sample is extremely limited, one humidity sensor is placed 1.0 cm over the PAAS sample while the other one is placed 30 cm away from the sample (**Fig. R7a**). These two sensors are placed at the same height above the wood board. It is shown in **Fig. R7b** that the RH over the PAAS sample is always higher than the ambient due to the occurrence of continuous evaporation. The average difference of the RH value is 6.5% which is really limited even from 10:00 to 14:00 when evaporation is strong during the day.

Figure R7: (a) Experimental setup to measure the RH of ambient and point over the PAAS photonic film during evaporation. (b) RH variation of sensor A and sensor B during the experimental period.

In order to thoroughly investigate the effect of humidity on our hybrid cooling material, we specifically selected a day with an ambient relative humidity of 50% RH or higher, with the average relative humidity measuring 53% RH from 10:00 to 20:00. For reference, we have presented the solar intensity and relative humidity data in **Fig. R8a**. As **Fig. R8b** illustrates, the radiative cooling performance fluctuated due to the change of the local environmental condition. However, the cooling performance of the hybrid mechanism is much more stable, further demonstrating the superior cooling performance of the proposed PAAS film. This exceptional performance was primarily attributed to the continuous process of evaporation from the top surface of the hybrid cooling material. Even in high-humidity environments, the evaporation process remained highly efficient, allowing our hybrid cooler to maintain its cooling capabilities and achieve significant sub-ambient temperatures. This outcome serves as a testament to the robustness and versatility of our cooling material, enabling it to adapt effectively to various climatic conditions, including humid climates.

Figure R8: (a) Solar intensity and relative humidity during the experimental period. (b) Temperature variation of the hybrid and radiative cooling samples during the experiment.

Reviewer #3:

This manuscript presents a comprehensive study on a hydrogel designed for atmospheric-moisture-induced evaporative and radiative cooling. The authors provide a robust background and detailed results, introducing a photonic PAAS that exhibits not only exceptional optical properties but also reusability and self-healing capabilities. I recommend the publication of this manuscript in Nature Communications, subject to minor revisions.

[Response]: We express our sincere appreciation to the reviewer for the positive evaluation of our study. The valuable feedback has been instrumental in enhancing the quality and clarity of our work. We have meticulously examined each of the suggestions and have taken them into serious consideration. Below, we provide detailed responses to address the points the reviewer raised:

1. In Figure 3e, the term 'wavelength' appears to be obscured by the figure below. Could the authors please rectify this?

[Response]: We appreciate the reviewer for bringing this to our attention. We have adjusted Figure 3e to ensure that the term “wavelength” is clearly visible and not obstructed by any elements in the figure.

2. The manuscript suggests that PAAS microstructures exhibit a scattering efficiency independent of diameter. This result seems counterintuitive. Could the authors please verify this and provide further clarification?

[Response]: We express our sincere appreciation for the insightful comments provided by the reviewer, and we humbly apologize for the error in our original algorithm for modeling the scattering efficiency of PAAS microstructures. In this revision, we have diligently rectified the modeling process, and the corrected results are now depicted in **Figure R9** (or Fig. 3e in the main text). The scattering efficiency should be dependent on the diameter of the microstructures.

Figure R9: Scattering efficiency plot of PAAS nano and microstructures.

To further validate the accuracy and reliability of this updated modeling approach, we employed this method to assess the scattering feature of nanoparticles and compared it with the published results extracted from ref R1, as evidenced in **Figure R10**. Through these corrective measures and additional validation

steps, we are confident that our scattering efficiency modeling now accurately represents the behavior of PAAS microstructures, ensuring the credibility and robustness of our research findings.

Figure R10: (a) Scattering efficiency contour plot of Cu nanoparticles that is extracted from **ref R1**. (b) Scattering efficiency plot simulated using our method.

Reference

R1. Tian, Y., Liu, X., Caratenuto, A., Li, J., Zhou, S., Ran, R., ... & Zheng, Y. (2022). A new strategy towards spectral selectivity: Selective leaching alloy to achieve selective plasmonic solar absorption and infrared suppression. *Nano Energy*, 92, 106717.

3. Figure 4b illustrates the cooling power and temperatures of ambient air and the PAAS film. To measure cooling power, heat must be applied to the PAAS film until it reaches ambient air temperature. Could the authors elaborate on the methodology used to measure this?

[Response]: We apologize for the lack of clarity caused by the representation of temperature reduction and cooling power in the same plot in Figure 4b. In our experiment (i.e., Figure 4b), we aimed to demonstrate the radiative cooling capability of the dry PAAS photonic film in terms of temperature reduction from ambient temperature. We performed experiments outdoors to measure the temperature reduction of the PAAS film in the PE film-covered chamber. By monitoring the temperature of the surface of the film over time, we observed a decrease in temperature compared to ambient air. Then, we estimated the radiative cooling power for this experimental data theoretically and plotted it in the same figure. However, we agree with the reviewer's point measuring cooling power typically involves heating the surface until it reaches the ambient air temperature and fluctuates with the varying ambient temperature. This process is employed by using a feedback control system, where the ambient air temperature is measured by the system and the feedback is utilized to match the temperature of the surface by heating.

In our specific case, due to the poor thermal conductivity of the dry PAAS film, it was challenging for us to directly measure the cooling power directly using the feedback control system as commonly employed. As a result, we opted for an alternative approach. Instead of measuring the cooling power directly, we calculated the radiative cooling power of the PAAS film based on experimentally observed temperature differences between the film and the ambient air. This estimation strategy is also widely used (e.g., ref. [R2-R4]) and was summarized in a recent comment reported by Nature Sustainability (i.e., ref. [R5]).

By plotting the radiative cooling power as a function of time in Figure 4b, we aimed to provide a quantitative assessment of the radiative cooling capability of the PAAS film. We acknowledge that this methodology differs from the conventional approach and should have been explicitly clarified in the manuscript.

In response to this valuable feedback, in this revision, we added clarity in the description of Figure 4b for the reader and address any confusion regarding the cooling power representation in Figure 4b. Finally, we provided a detailed description of the methodology employed to estimate the radiative cooling power (see supplementary notes S6).

Reference

R2. Wang, X., Liu, X., Li, Z., Zhang, H., Yang, Z., Zhou, H., & Fan, T. (2020). Scalable flexible hybrid membranes with photonic structures for daytime radiative cooling. *Advanced Functional Materials*, 30(5), 1907562.

R3. Li, J., Liang, Y., Li, W., Xu, N., Zhu, B., Wu, Z., ... & Zhu, J. (2022). Protecting ice from melting under sunlight via radiative cooling. *Science advances*, 8(6), eabj9756.

R4. Zhang, H., Ly, K. C., Liu, X., Chen, Z., Yan, M., Wu, Z., ... & Fan, T. (2020). Biologically inspired flexible photonic films for efficient passive radiative cooling. *Proceedings of the National Academy of Sciences*, 117(26), 14657-14666.

R5. Zhou, L., Yin, X., & Gan, Q. (2023). Best practices for radiative cooling. *Nature Sustainability*, 1-3.

4. The PAAS film appears to have a thickness exceeding 1 mm, which is considerably thicker than many radiative cooling films. The thickness of radiative cooling films can have both advantages and disadvantages. Could the authors discuss the impact of the thickness of their PAAS film on its performance?

[Response]: We sincerely appreciate the valuable feedback provided by the reviewer, particularly regarding the thickness of the PAAS film used in our study. It is indeed true that the PAAS film has a thickness exceeding 1 mm, making it thicker than many traditional radiative cooling films. We acknowledge that the thickness of the PAAS film plays a pivotal role in determining the overall performance of the cooling system, and it comes with its own set of advantages and disadvantages.

One notable disadvantage of thicker films is their potential to hinder the effective emission of thermal radiation from the underlying radiative cooling films. This could result in a decrease in the overall radiative cooling capacity, especially if the film absorbs and scatters a significant amount of the outgoing thermal radiation. However, there are several significant advantages associated with a thicker PAAS film:

The primary advantage is the increased moisture absorption capacity enabled by the greater thickness. A thicker film can store a larger volume of water (as demonstrated in **Fig. R11a**), facilitating prolonged and continuous evaporative cooling. This attribute proves especially valuable in arid or semi-arid climates, where water availability may be limited. Additionally, thicker PAAS films have a higher heat capacity, allowing them to store more thermal energy during the day. Consequently, the cooling effect can be sustained for extended periods, even after sunset. Moreover, increased thickness often imparts mechanical robustness and durability to the PAAS film, rendering it more resistant to physical damage and wear. This enhanced durability contributes to extending the operational lifespan of the cooling application.

In response to the impact of the PAAS film's thickness on its performance, we conducted an additional outdoor experiment in this revision. This experiment involved testing PAAS films with varying thicknesses

to assess their cooling performance, and we carefully monitored the solar intensity and relative humidity during the experimental period (as displayed in **Fig. R11b**). Our findings indicated that while a thicker hydrogel film indeed enhances moisture absorption and heat storage capabilities, it also introduces certain trade-offs, such as reduced radiative cooling efficiency. After careful evaluation, we determined that a 2-mm-thick film provided the optimal cooling performance (as shown in the temperature response graph in **Fig. R11c**). The PAAS photonic film with a thickness of 2 mm exhibited the lowest temperature during the experimental period. This particular thickness strikes a balance between moisture absorption capacity and radiative cooling efficiency, ultimately delivering the best overall cooling performance. Consequently, this thickness value is employed and discussed in our manuscript.

In this revision, we have added a section on thickness-dependent cooling performance in the supplementary information (on page 32) and discussed it in the main text (lines 390 to 393 on page 9).

Figure R11: (a) Water uptake of PAAS photonic film at different thicknesses. (b) The solar intensity and relative humidity during the experimental period. (c) Temperature variation of the hybrid and radiative cooling samples during the experiment.

REVIEWERS' COMMENTS

Reviewer #1 (Remarks to the Author):

The revised version is well-written based on all the reviewers' concerns.

Reviewer #2 (Remarks to the Author):

The authors have addressed all of my comments. The paper can be accepted

Reviewer #3 (Remarks to the Author):

I would like to publish this paper without changes. All concerns that I raised were addressed.